# End-to-End Detection of a Landing Platform for Offshore UAVs Based on a Multimodal Early Fusion Approach

**DOI:** 10.3390/s23052434

**Published:** 2023-02-22

**Authors:** Francisco Soares Neves, Rafael Marques Claro, Andry Maykol Pinto

**Affiliations:** 1Faculty of Engineering, University of Porto (FEUP), 4200-465 Porto, Portugal; 2Centre for Robotics and Autonomous Systems—INESC TEC, 4200-465 Porto, Portugal

**Keywords:** object detection, sensor fusion, early-fusion, computer vision, RGB camera, thermal camera, 3D LiDAR

## Abstract

A perception module is a vital component of a modern robotic system. Vision, radar, thermal, and LiDAR are the most common choices of sensors for environmental awareness. Relying on singular sources of information is prone to be affected by specific environmental conditions (e.g., visual cameras are affected by glary or dark environments). Thus, relying on different sensors is an essential step to introduce robustness against various environmental conditions. Hence, a perception system with sensor fusion capabilities produces the desired redundant and reliable awareness critical for real-world systems. This paper proposes a novel early fusion module that is reliable against individual cases of sensor failure when detecting an offshore maritime platform for UAV landing. The model explores the early fusion of a still unexplored combination of visual, infrared, and LiDAR modalities. The contribution is described by suggesting a simple methodology that intends to facilitate the training and inference of a lightweight state-of-the-art object detector. The early fusion based detector achieves solid detection recalls up to 99% for all cases of sensor failure and extreme weather conditions such as glary, dark, and foggy scenarios in fair real-time inference duration below 6 ms.

## 1. Introduction

Object detection is a perception task that provides the autonomous systems the necessary awareness of the surrounding environment. By extracting features from raw sensory information, it produces meaningful high-level knowledge about surrounding objects (e.g., other vehicles, pedestrians, goal marks, road lanes). In the end, object detection provides semantic representations about the surrounding environment [1]. For detection, the most common on-board sensors used by autonomous systems are two-dimensional/three-dimensional LiDAR, visual cameras, radar, and thermographic cameras. Each sensor has its own advantages and disadvantages that are directly affected by the environmental conditions. Visual cameras alone are the traditional sources of information, however, in recent years, there is a concern for diversifying the set of sensors to increase the robustness of the system for the case of sensor failure against different environmental conditions [2,3,4]. For example, thermal cameras discriminate really well different objects emitting different temperature intensities. However, on extreme heated environments (e.g., sunny days, hot rooms), the background and foreground becomes indistinguishable and homogeneous. Visual cameras provide colour and texture information, but for foggy, glary, or dark environments, the signal becomes corrupted and noisy. LiDAR sensors extract depth information by analyzing light energy reflected from objects in the surroundings. Nevertheless, in rainy and foggy environments, the reflected information becomes worthless. Hence, for an autonomous system to be reliable, it is a requirement to endow it with a diverse and distinct set of sensor equipment.

Adopting sensor fusion is one of the building blocks to build reliable perception systems. A system of this kind is still able to meaningfully perceive the environment, not only when there are individual failure, but also when environmental conditions corrupt any sensory information making it not reliable. At greater length, sensor fusion is described by combining the information produced by various sensors representing different modalities (e.g., visual, thermographic, RADAR, LiDAR) into a joint representation to produce a less uncertain input. Sensor fusion can happen in three different stages of the detection pipeline [5]: (i) early fusion refers to the combination of multiple input sources into a unique feature vector before feature extraction; (ii) intermediate fusion refers to the process of extracting correlation between features from the joint combination of sensory information inside the perception module and before classification or regression; and (iii) late fusion happens after the classification/regression procedure.

Extracting and combining relevant features from multiple modalities is a challenging process that requires a lot of domain knowledge and it strongly depends on the requirements of the task to solve. Effectively, deep learning technology is already capable of tackling this process [6,7,8] autonomously. Apart from this deep approaches, when significant correlation across modalities is smoothly inferred, early fusion should be considered [4]. If it is applied, early fusion produces early joint representations directly from raw data and anticipates correlation and redundancy. Several studies have been conducted to find several pair combinations between point cloud, visual, and thermographic into single concatenated representations [9,10,11,12]. However, there is a lack of research on combining the three.

In this paper, we contribute by presenting a novel multimodal early-fusion-based perception system that combines visual, thermal, and three-dimensional LiDAR data information to produce reliable detection capabilities against demanding operating conditions such as extreme weather or modality failure. According to the authors’ knowledge, there is no equivalent early-fusion method combining point clouds, thermographic, and visual information. This combination of sensors is implemented by a multimodal fiducial ArUco marker called ArTuga (Patent pending (Portuguese Patent Request (PPP) nr. 118328, and European Patent Request (EP) nr. 22212945.4)) proposed by Claro et al. [13]. It enables multimodal detection against several weather conditions for robotic solutions endowing heterogeneous perception systems comprised by visual, thermographic, and LiDAR-based devices. Therefore, it is not possible to replace the marker with an ordinary object given the constructive characteristics of the ArTuga that provide a spatial alignment of certain elements that facilitate the precise and robust detection of the marker. The contributions of this article include:Real-time multimodal marker detection that is deployable onboard a UAV;Resilient and high-accuracy detection based on early-fusion approaches against sensor failure integrated in the YOLOv7 framework;Robustness against demanding weather and operating conditions for extensive experiments using real UAVs landing on a floating platform;A new multimodal dataset collected in real offshore and onshore environments during a UAV landing operation, comprised of diverse joint representations of visual, infrared and point-cloud images.

The paper is organized as follows: In Section 2, there is a review of the literature about object detection and sensor fusion. In Section 3, a novel early-fusion methodology is proposed. Section 4 describes the experimental setup. Section 5 exposes the achieved results. Lastly, Section 6 provides the conclusions.

## 2. Related Work

Object detection (OD) is a computer vision task described by recognizing, identifying, and locating objects of interest within a picture with a certain degree of confidence. OD has gained popularity since it started to fuse with the strong capabilities of neural network technology. OD is mainly subdivided between single-stage and two-stage detection. These two categories compete by finding the optimal trade-off between detection speed and accuracy [14]. Real-time systems require fast and accurate predictions, therefore building real-time object detectors demands solving the former mentioned trade-off.

Two-stage detectors, such as the RCNN [15], fast-RCNN [16], and faster-RCNN [17], follow region proposal principles that focus on the localization of regions of interest in the image (e.g., where objects of interest are located) before performing detection. Firstly, these regions are estimated. Lastly, from these regions, detection is performed. Depending on the detected objects, correspondent confidence probabilities are extracted. These methods focus on achieving high detection accuracy, however, at the cost of being slow. Hence, two-stage detection is not viable for real-time demands. On the other hand, single-stage detectors, such as the you only look once (YOLO) [18] and the single-shot multibox detector (SSD) [19] perform detection and classification in a singular common step. Single-stage detection prioritizes inference speed over accuracy. Naturally, it is suitable to address real-time constraints. Despite inference speed prioritization, these methods achieve at least similar accuracy to two-staged detectors [14]. Within the available single-stage detectors, YOLO is a suitable choice: (i) it is more popular, accessible, and has a stronger documentation than any other method in the literature; (ii) it achieves highly fair detection speeds (up to 45 FPS), and thus it has at least similar accuracy when compared with other two-stage detectors.

Further, multimodal fusion [20] aims to find the combination of apparently disparate multi-domain data (e.g., visible light, infra-red light, sound, laser) sources to produce a more robust and more rich fused signal. One of the hot topics of the current sensor fusion literature is choosing the level in which the fusion takes place in the detection pipeline [5,21,22]. Sensor fusion happens in three levels, such as: (i) *early fusion* as the pixel-level fusion; (ii) *intermediate fusion* as the feature-level fusion; (iii) *late fusion* as the decision-level fusion. Several studies have been conducted by fusing sensory data in different levels of the perception pipeline. Farahnakian and Heikkonen (2020) [10] achieved state-of-the-art performance to detect marine vessels by applying intermediate fusion techniques using thermal and RGB cameras. However, reasonable real-time inference durations (mostly in the order of seconds) were not achieved. Additionally, the authors even suggested the possibility of adding a LiDAR source to the fused input to explore possible improvements. Liu et al. (2022) [12] proposed an intermediate fusion approach using LiDAR and visual camera data for car detection. Choi and Kim (2021) [11] followed an early fusion approach by combining an infrared camera with a three-dimensional LiDAR sensor, achieving reliable performance in impractical environmental conditions for vision-based sensors. Azam et al. (2019) [23] suggested another early fusion approach that fuses three-dimensional LiDAR with thermal images, ensuring reliable performance for both day and night light conditions. Bhanushali et al. (2020) [24] achieved reliable real-time object detection by training an end-to-end SSD detector on the KITTI dataset [25], merging early and late fusion principles. The former study also suggested the addition of other sensors, such as radar, to increase the robustness of the proposed model.

## 3. Multimodal Early Fusion Approach for Fiducial Marker Detection

Offshore robotic applications operate in challenging weather conditions where corrupt sensor information or even sensor failure situations are expected. Therefore, robotic solutions need to be resilient and endow redundant and heterogeneous perception systems [13]. Thus, based on the available multimodal sources, this research proposes a simple methodology that produces an expressive and redundant joint representation over a multimodal feature space. The proposed system comprises an (i) early-fusion module that applies a novel early-fusion technique and a (ii) lightweight YOLO-based detector that is fed with an early fused input. A system high-level perspective is depicted by the Figure 1.

### 3.1. Early Fuser

Based on spatiotemporal alignment of data streams, the early fuser produces a concatenated 3-channel RGB input containing a joint representation of visual, infrared, and LiDAR modalities. The aim is to anticipate redundancy across input streams and facilitate the detector’s feature extraction process. The early fuser procedure is described by: (i) a calibration step where modalities are aligned into a common coordinate system, (ii) a pre-processing step where relevant features for each modality are extracted, and (iii) a final concatenation step where processed modalities are aggregated into a redundant RGB representation.

#### 3.1.1. Calibration

To aggregate sensory information into a common three-channel image, spatial relationships between sensors must be obtained. The visual camera is chosen to be the main frame of reference. Hence, the remaining infrared and LiDAR information are projected into the visual camera image coordinate system. Before projecting the three-dimensional LiDAR point-cloud into the visual image coordinate system, the point-cloud is transformed into the visual camera coordinate system. The three-dimensional spatial relationship between each sensor is described by an extrinsic matrix. Using Zhang’s method [26], an extrinsic calibration operation is performed to obtain the extrinsic transformation matrix $E_{l,v}=\left[\begin{array}{cc} R & T \end{array}\right]$. $E_{l,v}$ is a rigid body transformation described by a rotation
$$R=\left[\begin{array}{ccc} r_{11} & r_{12} & r_{13}\\ r_{21} & r_{22} & r_{23}\\ r_{31} & r_{32} & r_{33} \end{array}\right]$$
and a translation $T={\left[\begin{array}{ccc} t_{x} & t_{y} & t_{z} \end{array}\right]}^{T}$ between the visual camera coordinate system *v* and the LiDAR coordinate system *l*, where LiDAR points $P_{l}={\left[\begin{array}{ccc} x_{l} & y_{l} & z_{l} \end{array}\right]}^{T}$ are converted into visual camera points $P_{v}={\left[\begin{array}{ccc} x_{v} & y_{v} & z_{v} \end{array}\right]}^{T}$ as follows:(1)$$\begin{array}{cc} P_{v} & =E_{l,v}P_{v}\\ \left[\begin{array}{c} x_{v}\\ y_{v}\\ z_{v} \end{array}\right] & =\left[\begin{array}{cccc} r_{11} & r_{12} & r_{13} & t_{x}\\ r_{21} & r_{22} & r_{23} & t_{y}\\ r_{31} & r_{32} & r_{33} & t_{z} \end{array}\right]\left[\begin{array}{c} x_{l}\\ y_{l}\\ z_{l} \end{array}\right]. \end{array}$$ Following, to transform $P_{v}$ into pixel points $U_{v}={\left[\begin{array}{ccc} u_{v} & v_{v} & 1 \end{array}\right]}^{T}\in {ℜ}^{2}$, an intrinsic calibration for the visual camera is performed using the Zhang’s method to obtain the intrinsic matrix defined by:$$K_{v}=\left[\begin{array}{ccc} f_{x} & \gamma & c_{x}\\ 0 & f_{y} & c_{y}\\ 0 & 0 & 1 \end{array}\right],$$
where $\left(f_{x},f_{y}\right)$ are the focal lengths, $\left(c_{x},c_{y}\right)$ is the image center and γ is the skew between *x* and *y* directions. Therefore, a perspective projection is applied to obtain $U_{v}$ as follows:$$\alpha \left[\begin{array}{c} u_{v}\\ v_{v}\\ 1 \end{array}\right]=K_{v}\left[\begin{array}{c} x_{v}\\ y_{v}\\ z_{v} \end{array}\right],$$ Finally, similar to a stereoscopic camera, an homografy matrix was extracted to map corresponding points between the thermal image and the visual image as carried out by [13].

#### 3.1.2. Pre-Processing

Specific pre-processing techniques are applied for each data stream. The visual stream is undistorted using OpenCV library, to remove natural lens distortion described by $K_{v}$. There is no need for rectification because it is a monocular camera. To be represented in a single-channel shape, the visual source, acquired in a RGB representation, is finally converted into a grayscale (single-channel) form. The infrared stream is already acquired in a grayscale representation. Similarly to the visual images, the infrared images undergo undistortion operations by resorting to the OpenCV library using lens distortion described by its intrinsic matrix.

The infrared sensor is properly pre-configured to capture distinct temperature contrast between the ArTuga and the background information. The temperature contrast results in a colour contrast. Availing this natural contrast, a binary threshold filter is applied as an enhancement operation and a filter operation to filter surrounding background noise. The binary operation is a pixel-wise non-linear threshold operation described by:(2)$$p_{ij}^{binarized}=\left\{\begin{array}{cc} 255 & p_{ij}^{orig}>thresh\\ 0 & \mathrm{otherwise} \end{array}\right.,$$
where 255 is the maximum color value, thresh=128 is the threshold applied, $p_{i,j}^{orig}$ is any original pixel and $p_{i,j}^{binarized}$ is any transformed pixel. The binary threshold operation is depicted in the Figure 2.

As it can be observed, the binarization results in a more recognizable object by enhancing colour contrast and cleaning the background while preserving edge and corner properties.

The LiDAR stream is acquired in a three-dimensional point-cloud representation. Before intrinsic/extrinsic transformations, this stream is pre-processed to filter and enhance specific inherent information. Excessive and needless point-cloud information is filtered using a voxel downsampling operation. Voxel downsampling is a spatial operation that iteratively buckets points into three-dimensional voxels. Each voxel is compressed and generates a unique three-dimensional point by averaging every inner point. This operation reduces the size of the cloud and throws out excessive information while retaining the overall geometric structure. The used voxels have a leaf size of 5 cm. In addition, more points are removed according to their intensity. The ArTuga has a retro-reflective tape in its borders and white-bit coding area in its interior resulting in disparate values of intensity in comparison to the rest of the point cloud. Points with intensity below a maximum intensity of *I* are suppressed. Hence, the resulting points mostly belong to the aruco object. A two-dimensional grayscale image is generated by assigning a colour to each resultant pixel, depending on the laser depth, according to a 8-bit scaling operation. The colour *c* for each pixel is computed as follows:$$c=255\left(1-\frac{d}{m_{d}}\right),$$
where *d* is the laser distance measurement and $m_{d}$ is the maximum depth reached during the UAV flight. A final dilation operation is applied to compensate the sparsity of the image, by applying a 30×30 squared kernel as depicted by the Figure 3.

#### 3.1.3. Concatenation

As a final step, the pre-processed single-channel streams are concatenated into a three-channel RGB image, where the red, green, and blue channels correspond to the LiDAR, infrared, and visual streams, respectively. The channels are presented in the Figure 4.

The concatenation finalizes the early fusion pipeline resulting in an input that is more understandable and correlated across modalities to feed the YOLO detector. The pipeline is depicted in the Figure 5.

### 3.2. Detector

Nowadays YOLO-based detectors are a solid and popular choice for fast but still accurate detection. Impressively, it is used to tackle real-world tasks (e.g., autonomous cars and UAVs) [27,28,29], addressing challenging weather conditions [30,31] and real-time speed constraints [32,33]. Nowadays, the most recent versions of YOLO, the YOLOv5 [34], YOLOv6 [35], and the YOLOv7 [36] are the state-of-the-art detectors for fast and accurate detection. This work uses a detector based on the YOLOv7 architecture. Considering real-time demands, the feasibility of the smallest version of the YOLOv7 is explored: the *tiny* version. The aim is to scrutinize the feasibility of these lightweight, faster, and simpler detectors considering the weaker accuracies when compared to the bigger versions. Hence, the proposed early fusion method aims to counteract lightweight limitations by facilitating the input information.

More particularly, YOLO addresses the detection task as a regression problem because it outputs bounding box coordinates (location of objects in the image) and the probability of detection (the confidence in the prediction) for each detected object. YOLO takes a full image as an input, diving it into a into a n×n grid. For each cell in the grid, it estimates *N* possible bounding boxes correspondent to *N* objects. Each detected object is classified by a label *l* (e.g., l∈{pedestrian,truck,traffic light}). Each bounding box *b* is defined as a vector as follows:$$b=\left[\begin{array}{ccccc} x_{min} & \;y_{min} & \;x_{max} & \;y_{max} & \;c\;l \end{array}\right],$$
where $(x_{min}$, $y_{min})$ and $(x_{max}$, $y_{max})$ are the minimum and maximum pixel coordinates, respectively, with respect to the top-left corner of the image, *c* is the confidence of the prediction containing an object and *l* is the object class label. Ultimately, YOLO outputs a N×6 dimensional tensor, where only the most trusted predicted bounding boxes remain, by applying a technique called Non-Maximum-Suppression.

## 4. Experimental Setup

A UAV named CROW (copter robot for offshore wind-farms), based on a quadcopter frame (with a wingspan of 0.7 m, and a maximum payload of 2 Kg), was remotely operated to execute a few minutes duration flight around a maritime platform that contains the ArTuga marker in its center. The CROW endows a perception system comprised of a (i) three-dimensional LiDAR, a (ii) visual camera, and an (iii) infrared camera. The detection performance is highly dependent on the sensor characteristics. Thus, the sensor choice must provide high resolution capabilities. The set of sensors have the following specifications:Visual Camera—*The Imaging Source DFM 37UX273-ML*—Frame Rate: 15 Hz, Resolution: 1440 × 1080 pixels, Field of View: 45° horizontal;Thermal Camera—*FLIR Boson 640 Radiometry*—Frame Rate: 15 Hz, Resolution: 640 × 512 pixels, Field of View: 50° horizontal, Temperature Measurement Accuracy: ±5 °C;3D LiDAR—*Ouster OS1-64*—Frame Rate: 10 Hz, Resolution: 64 × 1024 channels, Range: 120 m, Accuracy: ±0.05 m, Field of View: 360° horizontal and 45° vertical.

During the flight, the sensor data is recorded to be further processed to generate the datasets for model training. There was a concern to produce a heterogeneous dataset, therefore the flight operation was conducted considering different spatial perspectives of the platform. Despite the aim of this application being offshore, onshore samples were also acquired to promote heterogeneity. The high variability of an heterogeneous dataset ensures that a model is robust against unexpected and different image perspectives. Naturally, robustness is a major priority to avoid overfitting while training. The CROW UAV executing a landing operation is depicted by the Figure 6.

### 4.1. Datasets

The performance of the early fusion detector is going to be compared with three unimodal detectors correspondent to each sensor stream. The unimodal detectors function as fine-tuned baseline detectors for each modality. Hence, the aim is to use them as a performance reference for the multimodal early fusion detector. Accordingly four datasets are generated: a visual, a thermal, a LiDAR and a fusion datasets. The datasets are available in a Google Drive public repository [37].

#### 4.1.1. Visual Dataset

The visual dataset comprises 1449 images representing onshore and offshore visual samples from different spatial points of view of the ArTuga with different backgrounds (e.g., landing platform, water, ground). Besides the inherent variability of the data, some data augmentation techniques are applied for this dataset to prevent overfitness. Brightness variation is a clear augmentation technique to train the model against brighter and dark scenarios. The motivation for this technique is to prepare the detector for sunny and night settings. In more depth, for every image, by increasing up to 25% or decreasing up to 90% the brightness, a darker image is a pixel-wise operation, such as $p_{i,j}^{dark}=p_{i,j}^{orig}\left(1-\delta \right),\;\forall i\in \left[0,h-1\right]\subset N,\;j\in \left[0,w-1\right]\subset N$ and a brighter image such as $p_{i,j}^{bright}=p_{i,j}^{orig}\left(1+\beta \right),\;\forall i\in \left[0,h-1\right]\subset N,\;j\in \left[0,w-1\right]\subset N$, where δ∈]0,0.9], β∈]0,0.25], w,h are the image width and height, respectively, $p_{i,j}\in \left[0,255\right]\subset N$ is a pixel in the ith row and jth column of the image, and 0 and 255 are the minimum and maximum colour values. In addition, flipping and rotation augmentation techniques are applied. The images are resized from the acquisition size 1440×1080 to a final size of 640×480 pixels. The Figure 7 depicts some samples from the visual dataset.

#### 4.1.2. Thermal Dataset

The thermal dataset comprises 441 pre-processed binarized infrared images as already described in the Section 3.1.2. The data augmentation techniques applied for this dataset are flipping and rotation. The images are resized from the acquisition size 1440×1080 to a final size of 640×480 pixels. The Figure 8 depicts some samples from the thermal dataset.

#### 4.1.3. LiDAR Dataset

The LiDAR dataset comprises 316 pre-processed two-dimensional point cloud projections already described in the Section 3.1.2. The data augmentation techniques applied for this dataset are flipping and rotation. The images are resized from 1440×1080 to a final size of 640×480 pixels. Figure 9 depicts some samples from the LiDAR dataset.

#### 4.1.4. Fusion Dataset

The fusion dataset is constituted by 2158 pre-processed and concatenated RGB images produced by the early fuser described in the Section 3.1. This dataset implements a brightness augmentation technique for the visual channel and flipping and rotation techniques for all channels. The images are resized from 1440×1080 to a final size of 640×480 pixels. The Figure 10 depicts some samples from the the fusion dataset. As it can be observed, the dataset contains both multimodal samples representing cases when all modalities are available; and unimodal samples representing sensory failure cases when specific modalities are deactivated. Training with unimodal samples intends to promote resilience against sensory failure.

#### 4.1.5. Annotation

For each sample in the dataset, there is a correspondent annotation. Manual annotations in the form of a bounding box are performed using roboflow framework [38]. For the unimodal datasets, the bounding box fully encloses the unique object present as shown in the Figure 11a–c. In this way, it is noted that there is a care for pixel tightness to drive the model to the best accuracy. As for the case of the fused sample, the annotation considers the stream with the best resolution (shape and colour), as shown in Figure 11d. Additionally, negative samples are present in the dataset to teach the model when an object is not present as shown in the Figure 11e or partially present as depicted by the Figure 11f. For these negative cases, the annotation is not performed, instead it is marked as *null*.

## 5. Results

This section exposes (i) the training settings and results, (ii) an ablation test to evaluate the model’s resilience during landing for all cases of sensor failure, and (iii) a resilience test against challenging weather conditions for fog, dark, and glary scenarios.

### 5.1. Training

#### 5.1.1. Training Settings

The training hyperparameters were tuned according to useful guidelines towards reliable training performance provided by the official documentation of the YOLO framework (https://github.com/ultralytics/yolov5/wiki/Train-Custom-Data—Accessed on 19 February 2023). The training procedure is executed for 400 epochs. The selected batch size is 32. The software training platform resorts to Google Colab’s servers (https://colab.research.google.com/—Accessed on 19 February 2023), having access to its free of charge NVIDIA Tesla T4 GPUs. An IoU threshold of 0.2 is chosen. A learning rate cosine scheduling [39] is applied with an initial learning rate of 0.01. An Adam optimizer is used because over SGD and RMS Prop it had the best mAP@0.5:0.95 performance. To prevent overfitting, data augmentation techniques inherently exist in the dataset, and a weight decay of $5\times 10^{-4}$ is applied. The training hyperparameters are presented in Table 1.

#### 5.1.2. Training Results

The training results for all the detectors are presented in the Figure 12. Commonly, a clear training convergence is observed for all models because the train regression loss (train/box_loss) and the objectness loss (train/obj_loss) monotonically decrease. The *box* loss describes the decrease in the regression error between the predicted and ground-truth bounding boxes (x,y,w,h) values. The *obj* loss describes the error between the confidence the model has on the object presence and the true presence. A *cls* loss has a particular behaviour because this is a single-class problem: when there is a class, it always the class 0. Therefore the problem reduces to predicting the presence of an object and the better enclosing bounding box. In addition, the model does not overfit because the validation losses monotonically decrease and eventually settle into a final value.

Regarding accuracy, all models achieved high levels of accuracy (above 95%) both on the The mAP@0.5 and The mAP@0.5:0.95 metrics (above 70%). The mAP@0.5:0.95 is significantly lower because it is more demanding. The early fusion model has the lower accuracy on the mAP@0.5:0.95 metric because of the redundancy of the predictions. Due to being a multimodal detector there is always some uncertainty about the true location of the object because the modalities are not perfectly aligned. This misalignment between modalities is caused by the temporal asynchronism between sensors. Hence, on average, the accuracy is always lower when compared to the accuracy achieved by unambiguous unimodal detection. The precision and recall metrics also demonstrate the accuracy convergence across all models. The precision describes the validity of true positive prediction on the universe of the selected true positive labels. For this metric, there is a general convergence towards 1, despite the thermal detector. On the thermal samples, due to the non-linearity of the binarization operation, some detail is occasionally lost which negatively influences the prediction.

Similarly, for the recall metric, it is clear to observe a convergence towards 1, demonstrating that ultimately the predictions are complete: on the universe of all the images in an epoch containing a marker, the model accurately recognized a marker. The early fusion model slightly loses on the recall. This could be caused by the misalignment of some samples. When this misalignment is pronounced, the redundancy is lost and the model produces false negative predictions. This limitation could be avoided by including more unimodal samples on the early fusion dataset, such that the model trains more on unimodal situations (when the redundancy is not present).

Finally, to evaluate the performance of the detectors outside the training domain we compare the mAP score obtained by inferring the correspondent test datasets. Table 2 exposes the mean average precision for each detector.

All detectors achieve outstanding generalizable behaviour. Comparatively, the thermal detector stands out slightly. The thermal information, specifically, discriminates the object better which facilitates detection. Apart from that, thermographic information alone could suffer from particular extreme heat weather conditions (e.g., hot and sunny days). Thus, visual and point cloud information can complement this limitation. Decently, the early fusion detector nearly matches the fine-tuned detectors.

Lastly, the generalization capability of the detector is evaluated by inferring an external dataset called TNO Image Fusion [40] comprised by fused multispectral images as depicted by the Figure 13. Since this dataset does not contain the ArTuga marker, the aim is to examine the resilience of the detector against false positive predictions. Table 3 exposes the number of false positive predictions for five different levels of confidence thresholds across 127 images.

From Table 3, it is clear that below 75% the model starts to become sensitive. However, in a real application if the threshold is set above 80%, the model can be considered reliable. In conclusion, the results from this evaluation are substantial since they demonstrate the resilience of the detector against novel and noisy information.

### 5.2. Testing

The testing phase is comprised by an ablation and a weather tests to evaluate the model’s resilience against sensor failure and challenging weather conditions, respectively. Glary, dark, and foggy weather conditions are addressed. Particularly, it is not feasible to operate current UAV technology in rainy conditions, hence rain settings are not addressed. Additionally, it is inopportune to apply LiDAR technology for rainy conditions [41]. The hardware setup for testing comprises an Intel^®^ Core™ i7-10700F CPU @ 2.90GHz × 16 processor and a NVIDIA GeForce RTX 3060 GPU.

#### 5.2.1. Ablation Test

To evaluate the resilience of the proposed early-fusion detector, we perform an ablation study where sensor failure is simulated by intentionally deactivating several combinations of specific modalities during a UAV landing (from an high to a low altitude). Several cases of unable (deactivated) modalities are simulated as follows: only LiDAR, only thermal, only visual, LiDAR and thermal, LiDAR and visual, thermal and visual. As a reference all cases are compared to the baseline (None) where all streams are activated. For each case, 18 images, acquired during landing, are inferred to the model. Recall and inference time results are exposed in the Table 4. True positive predictions are considered for confidences above 80%. For evaluating the accuracy only the recall metric is considered since all the images have a marker present. Moreover, Figure 14 depicts the cases and the predictions.

The so needed redundancy against sensor failure is clearly concluded from the results exposed. Solid performance is demonstrated by the Recall results equaling 1 across all unable signal cases. This robustness is expected considering the presence of several unimodal samples in the dataset. Training with unimodal samples prepares the model against unable signals. Summing up, it can be concluded the reliability of the early fusion approach while operating under a sensor failure situation.

#### 5.2.2. Weather test

To motivate the use of an early-fusion based detector for challenging weather conditions a stress test for extreme simulated weather conditions is conducted. Glary and dark environments are simulated for increasing and decreasing, respectively, variations in brightness of 10%, 50% and 90% of the original image. Fog environments are simulated by applying stochastic fog augmentation using the Image Augmentation library (https://imgaug.readthedocs.io/en/latest/source/api_augmenters_weather.html—Accessed on 19 February 2023).

Furthermore, every prediction above a 0.5 confidence threshold is considered a true positive. Otherwise, it is considered a false negative. An amount of 100 images are inferred for testing. For every image, there is a marker present, thus, at best, the model should produce 100 true positive predictions. For evaluating the accuracy only the recall metric is considered since all the images have a marker present. The results achieved are presented in the Table 5. Examples of the extreme scenarios are depicted by the Figure 15.

It can be concluded that the model is more sensitive under extreme bright conditions. Conversely, it is resilient against dark and fog conditions. Considering the rare occurrence of extreme brightness situations on the dataset, the model naturally performs poorly against it. Resilience is expected against dark conditions because the dataset includes unimodal samples that mimic sensor failure. Concretely, the model is trained against unable visual signals or, in other words, extreme dark environments. The worthy performance (96% recall) against the stochasticity of fog conditions is the most surprisingly result, which demonstrates the robustness against a more noisy and random signal. Hence, it can be said that the model is presumably robust against real noisy environments.

## 6. Conclusions

This research introduced a novel methodology that proposes an early-fusion module capable of introducing the required reliability to a lightweight state-of-the-art object detector under sensor failure and extreme weather conditions.

The proposed early fusion approach provided an expressive input that clearly facilitated the detection of a multimodal fiducial marker during a UAV offshore operation. Together, the early fusion detector and the multimodal marker operate in a robust and transparent fashion against challenging weather and sensor failure conditions. In addition, if there is a GPU onboard a robotic solution, it should be emphasized the assurance of a fast system for real-time operating conditions, as demonstrated by the inference time results of less than 6 ms. Consequently, it can be said that assuring robust performance empowered by an early fusion and end-to-end approach for a lightweight detector is one of the major contributions of this work.

## Figures and Tables

**Figure 1 sensors-23-02434-f001:**
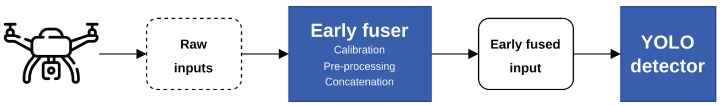
The early-fusion detection system high-level view.

**Figure 2 sensors-23-02434-f002:**
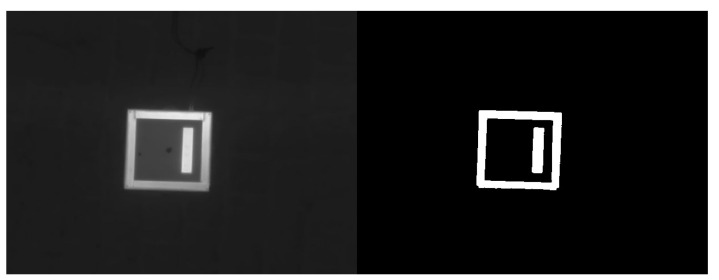
The binary threshold operation applied to the infrared images.

**Figure 3 sensors-23-02434-f003:**
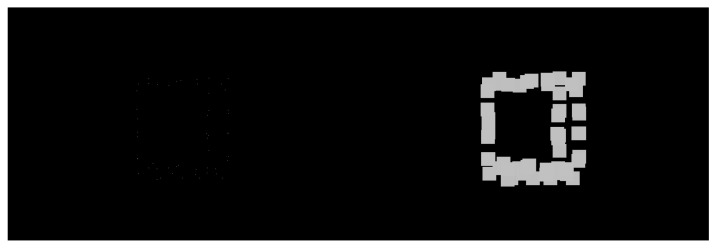
Dilation over the sparsed 2D points.

**Figure 4 sensors-23-02434-f004:**
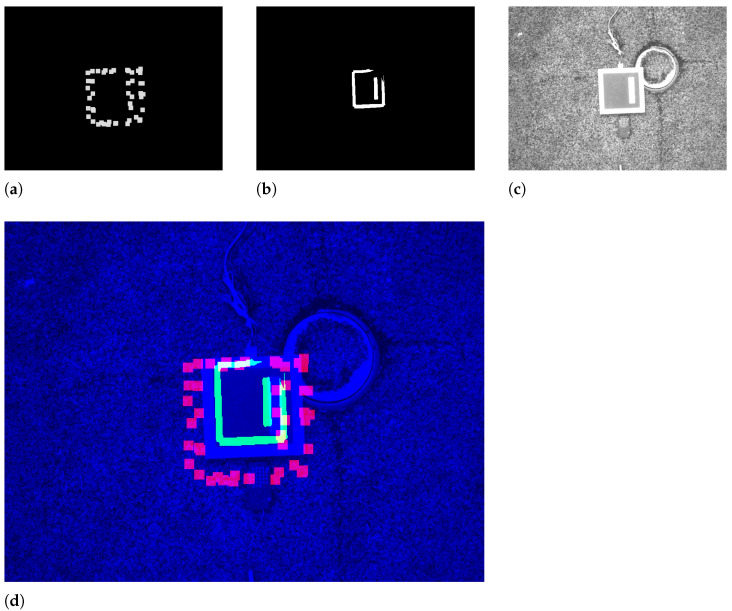
The individual channels and the aggregated RGB image. (**a**) The LiDAR channel. (**b**) The thermal channel. (**c**) The visual channel. (**d**) The aggregated image.

**Figure 5 sensors-23-02434-f005:**
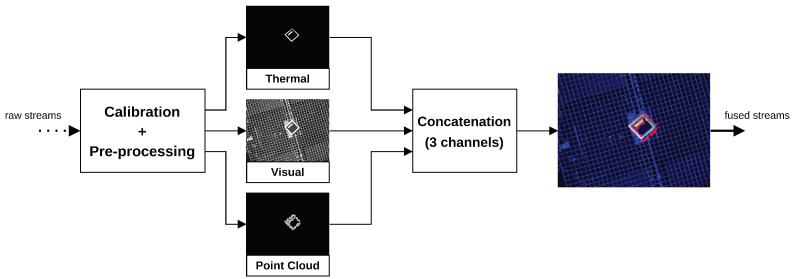
The early fuser pipeline.

**Figure 6 sensors-23-02434-f006:**
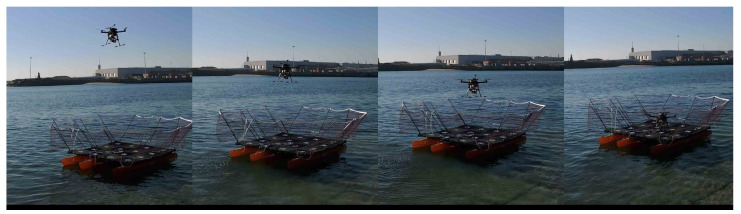
The conducted real experiment of a landing procedure using the CROW UAV.

**Figure 7 sensors-23-02434-f007:**
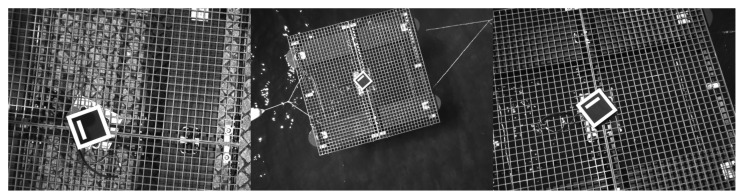
Samples from the visual dataset.

**Figure 8 sensors-23-02434-f008:**
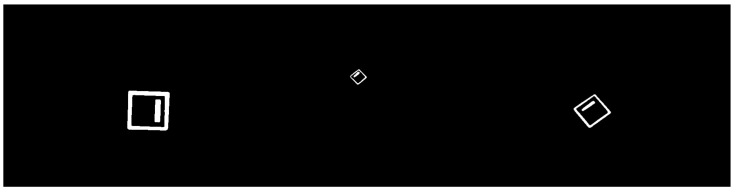
Samples from the thermal dataset.

**Figure 9 sensors-23-02434-f009:**
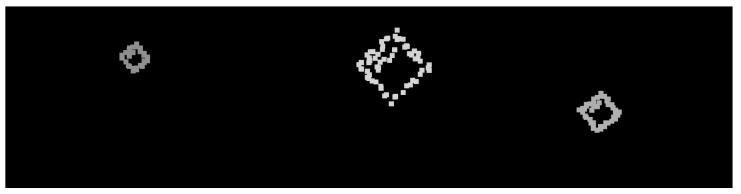
Samples from the LiDAR dataset.

**Figure 10 sensors-23-02434-f010:**
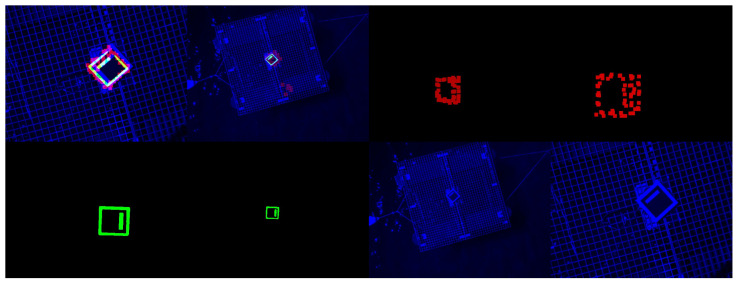
Samples from the Early Fusion dataset.

**Figure 11 sensors-23-02434-f011:**
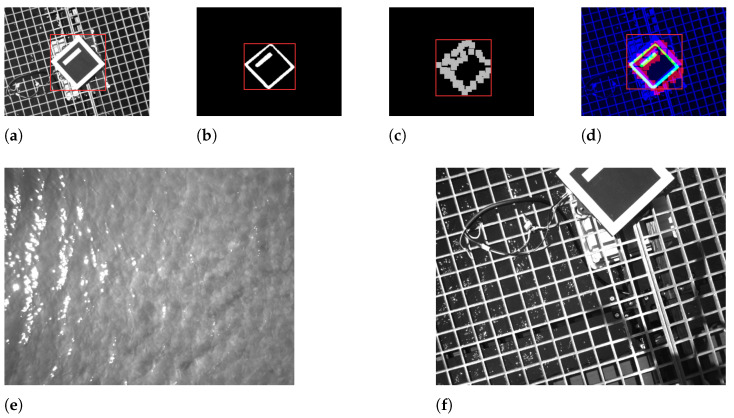
Non-null annotations on the top and null annotations on the bottom. (**a**) A visual sample annotation. (**b**) A thermal sample annotation. (**c**) A LiDAR sample annotation. (**d**) A fused sample annotation. (**e**) A null example (no object present). (**f**) A null example (partially present object).

**Figure 12 sensors-23-02434-f012:**
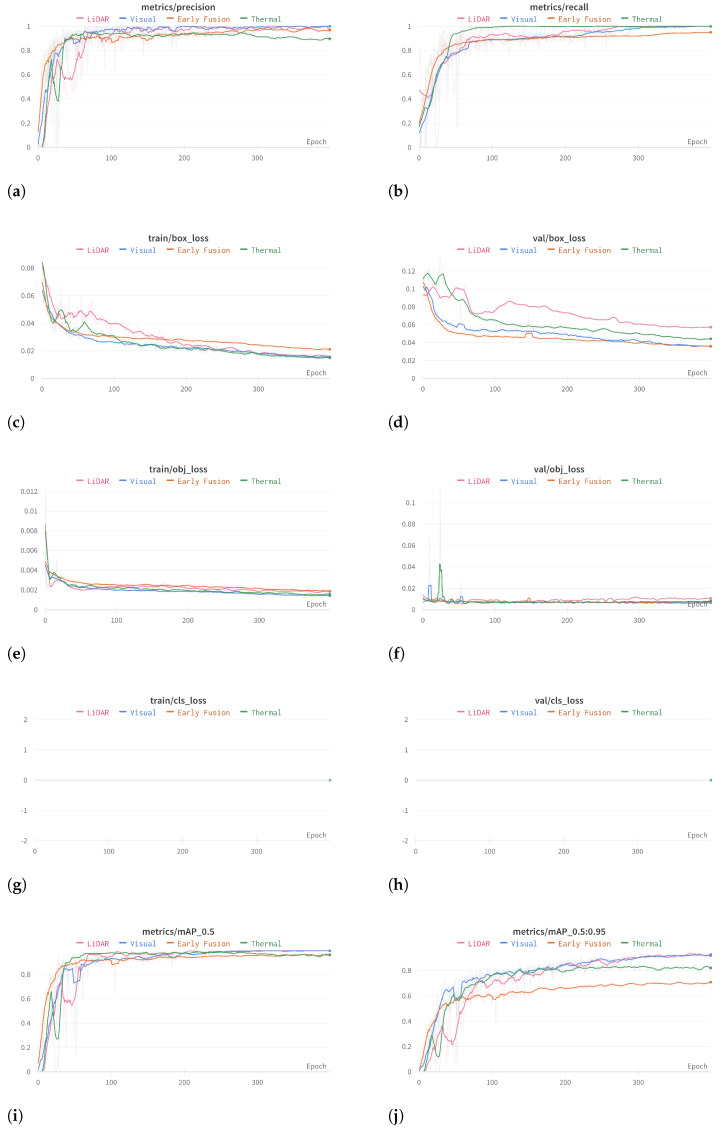
Training results. (**a**) The precision. (**b**) The recall. (**c**) The train regression box loss. (**d**) The validation regression box loss. (**e**) The train regression object loss. (**f**) The validation regression object loss. (**g**) The train regression class loss. (**h**) The validation regression class loss. (**i**) The Mean Average Precision 0.5 accuracy. (**j**) The Mean Average Precision 0.5:0.95 accuracy.

**Figure 13 sensors-23-02434-f013:**
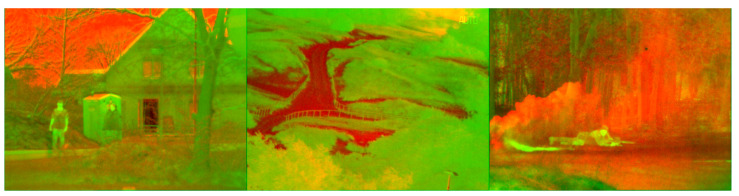
Some images from the TNO dataset [40].

**Figure 14 sensors-23-02434-f014:**
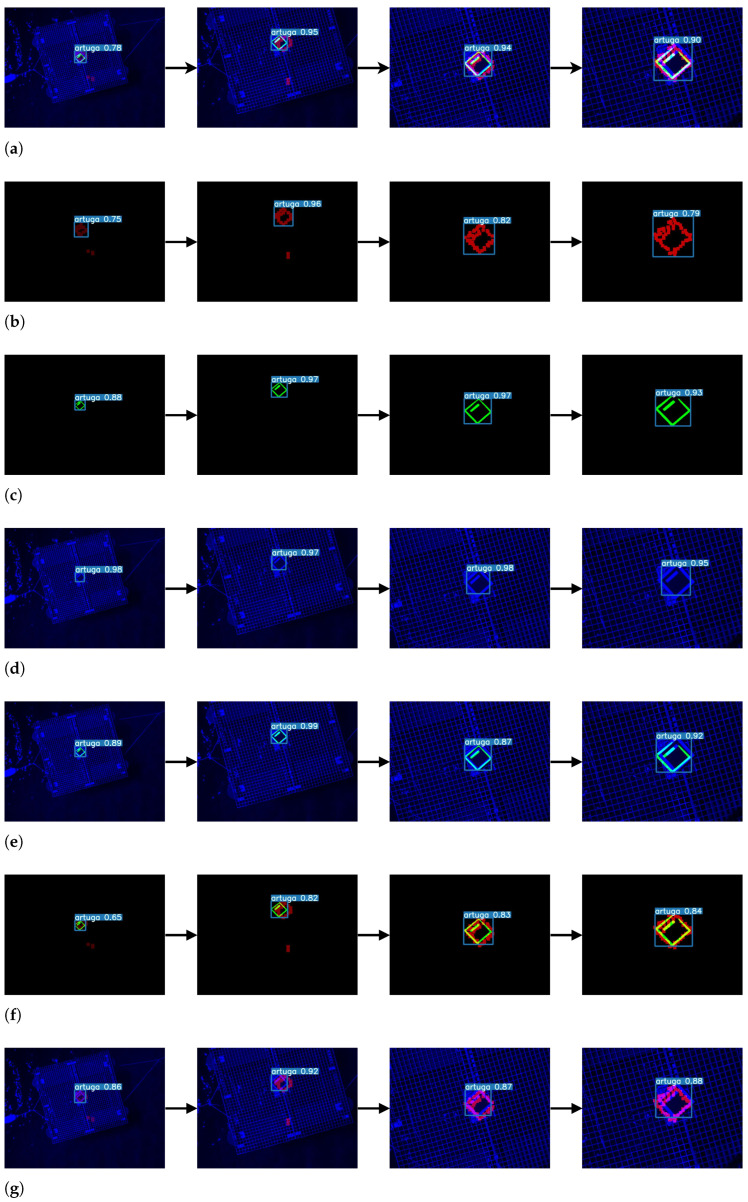
Examples of ablation test predictions for all sensor failure cases. (**a**) Early-fusion (baseline). All sources are activated. (**b**) Relying on LiDAR source only. (**c**) Relying on thermal source only. (**d**) Relying on visual source only. (**e**) LiDAR source failure. (**f**) Visual source failure. (**g**) Thermal source failure.

**Figure 15 sensors-23-02434-f015:**
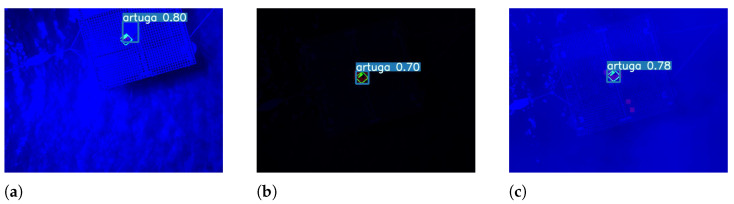
Examples of predictions under extreme weather conditions. (**a**) A 90% glary (sunny) scenario example. (**b**) A 90% dark (night) scenario example. (**c**) A fog scenario example.

**Table 1 sensors-23-02434-t001:** The main training hyperparameters.

Parameter	Quantity
Epochs	400
Batch size	32
IoU threshold	0.2
Momentum	0.937
Learning rate	$1\times 10^{-2}$
Weight decay	$5\times 10^{-4}$

**Table 2 sensors-23-02434-t002:** The performance of each detector on its own test set.

Detector	Modality	mAP@.5	mAP@.5:.95	Precision	Recall
Visual	Unimodal	0.999	0.999	1	1
Thermal	Unimodal	1	1	1	1
LiDAR	Unimodal	0.999	0.999	1	1
Early Fusion	Multimodal	0.989	0.989	1	0.985

**Table 3 sensors-23-02434-t003:** False positive evaluation on an external dataset.

Confidence Threshold (%)	False Positives
60	19
70	9
75	4
80	1
90	0

**Table 4 sensors-23-02434-t004:** Performance over 18 images during a landing operation for different unable signals.

Unable Signals	True Positives (Conf > 0.8)	Recall	Inference Time (ms)
None	100%	1	5.6
Visual and Thermal	100%	1	5.5
Visual and LiDAR	100%	1	4.2
Thermal and LiDAR	100%	1	4.7
LiDAR	100%	1	4.2
Visual	100%	1	4.1
Thermal	100%	1	4.8

**Table 5 sensors-23-02434-t005:** Performance over 100 images described by challenging weather conditions.

Weather Condition	True Positives (Conf > 0.6)	False Negatives	Recall	Inference Time (ms)
Bright (10%)	99	1	0.99	2.2
Bright (50%)	98	2	0.98	2.1
Bright (90%)	87	13	0.87	2.0
Dark (10%)	99	1	0.99	2.6
Dark (50%)	100	0	1	2.1
Dark (90%)	96	4	0.96	2.0
Fog (stochastic)	96	4	0.96	2.0

## Data Availability

The data produced by this research is described by four datasets designed to train Early Fusion Multimodal fiducial marker detectors for UAV landing operations. It is available at [37].

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
