# Peer review of "End-to-End Detection of a Landing Platform for Offshore UAVs Based on a Multimodal Early Fusion Approach"

_sensors, 2023, doi:10.3390/s23052434_

Round 1

Reviewer 1 Report

The paper is well written. I have the following comments.

(1)Is any approach exists to perform a similar early fusion approach? If exist then compare one such approach with your method. I think validation can be improved by comparing the proposed method with more detectors. If you find in literature then compare with one/two more detectors.

(2)TNO dataset is one of the widely used infrared-visible image fusion data set, which is readily available. Include the TNO dataset to your experiments. The corresponding article is:

(a)Toet, A. (2017). The TNO multiband image data collection. Data in brief, 15, 249-251.

(3) I think the Patent section, that is present as Section 7, is inappropriate here. REmove from that place. You may indicate this in Acknowledgement or like that. Or in Funding.

(4)What are unable signals? It is not much clear. Explain in a more detailed manner. 

(5)You may include the following articles to your literature.

(a)Yi, S., Jiang, G., Liu, X., Li, J., & Chen, L. (2022). TCPMFNet: An infrared and visible image fusion network with composite auto encoder and transformer–convolutional parallel mixed fusion strategy. Infrared Physics & Technology, 127, 104405.

(b)Panigrahy, C., Seal, A., & Mahato, N. K. (2022). Parameter adaptive unit-linking dual-channel PCNN based infrared and visible image fusion. Neurocomputing, 514, 21-38.

(6)Include more recent papers in the Introduction section and rewrite it.

(7) Check the whole manuscript for any grammatical/typo errors and remove short paragraphs.

Author Response

(1)Is any approach exists to perform a similar early fusion approach? If exist then compare one such approach with your method. I think validation can be improved by comparing the proposed method with more detectors. If you find in literature then compare with one/two more detectors.

Considering the authors knowledge there is no equivalent method (early fusion for multimodal data sources) capturing the combination of point clouds, visual and thermographic images.

(2) TNO dataset is one of the widely used infrared-visible image fusion data set, which is readily available. Include the TNO dataset to your experiments. The corresponding article is: (a) Toet, A. (2017). The TNO multiband image data collection. Data in brief, 15, 249-251.

The dataset was evaluated despite not including the multimodal fiducial ArTuga marker. We decided that this dataset could be useful to evaluate the resilience of our model against novel data. We could study the robustness of our model against possible false positive predictions.

This study was included in the end of the Training section which you can find in the new article version: "Lastly, the generalization capability of the detector is evaluated by inferring an external dataset called TNO Image Fusion \cite{Toet2022_dataset_tno} comprised by fused multispectral images as depicted by the Figure \ref{fig:tno_samples}. Since this dataset does not contain the ArTuga marker, the aim is to examine the resilience of the detector against false positive predictions. Table \ref{tab:tno_test} exposes the number of false positive predictions for five different levels of confidence thresholds across 127 images."

(3) I think the Patent section, that is present as Section 7, is inappropriate here. REmove from that place. You may indicate this in Acknowledgement or like that. Or in Funding.

I deleted the Patent section, and moved the text to the funding section that had already some content. You can find the update in the new uploaded manuscript version.

(4) What are unable signals? It is not much clear. Explain in a more detailed manner.

The text was rewritten as follows: "Several cases of unable (deactivated) modalities are simulated as follows: only LiDAR, only thermal, only visual, LiDAR and thermal, LiDAR and visual, thermal and visual. As a reference all cases are compared to the baseline (None) where all streams are activated."

(5) You may include the following articles to your literature. (a)Yi, S., Jiang, G., Liu, X., Li, J., & Chen, L. (2022). TCPMFNet: An infrared and visible image fusion network with composite auto encoder and transformer–convolutional parallel mixed fusion strategy. Infrared Physics & Technology, 127, 104405. (b)Panigrahy, C., Seal, A., & Mahato, N. K. (2022). Parameter adaptive unit-linking dual-channel PCNN based infrared and visible image fusion. Neurocomputing, 514, 21-38.

The suggested articles propose a pulse-coupled and an auto-encoder based network to interpret visible and thermographic information. They were referenced in the scope of the SOTA of Deep Learning which you can find in the Introduction section. Thank you for the suggestions.

(6) Include more recent papers in the Introduction section and rewrite it.

The introduction was reorganized and more recent papers were introduced. Thank you for the suggestion.

(7) Check the whole manuscript for any grammatical/typo errors and remove short paragraphs.

The article was properly reviewed by the authors to address eventual typos and short paragraphs.

Reviewer 2 Report

This paper investigated a detection pipeline of a landing platform for offshore UAVs based on a multimodal early fusion approach. The idea sounds interesting and the method looks applicable. To further improve the paper standard, my comments are:

1) I think the detection accuracy is highly related to the height of the UAV due to the multi-scale of the objects [1]. In the experiment, authors may further clarify these settings in the constructed dataset.  

2) from table 2, we can see that the detector based on thermal images gave the best results. So, are other modalities still useful for real-world applications? I think it is reasonable as vision-based detection methods still have unstable issue. Authors can further explain and clarify this.

3) in the weather testing, it would be better to test other weather conditions, such as rainy conditions, I hate to ask this though. If it is difficult, authors are suggested to discuss this or leave it to future work.

4) the last concern is about the designed marker. I am curious about this specific design. Authors may further explain this. Is it possible to replace this marker with some ordinary objects? 

Ref:

[1] Zhang, H., Sun, M., Li, Q., Liu, L., Liu, M. and Ji, Y., 2021. An empirical study of multi-scale object detection in high resolution UAV images. Neurocomputing421, pp.173-182.

Author Response

1) I think the detection accuracy is highly related to the height of the UAV due to the multi-scale of the objects [1]. In the experiment, authors may further clarify these settings in the constructed dataset. 

We would say, really, that the detection accuracy is more highly dependent on the sensor characteristics. And thus, the accuracy benefits as strongly as the provided sensory resolution. Proper additional information was included in the article in the Experimental Setup section as follows: "The CROW endows a perception system comprised of a (i) 3D LiDAR, a (ii) visual camera and an (iii) infrared camera. The detection performance is highly dependent on the sensor characteristics. And thus, the sensor choice must provide high resolution capabilities. The set of sensors have the following specifications..."

2) from table 2, we can see that the detector based on thermal images gave the best results. So, are other modalities still useful for real-world applications? I think it is reasonable as vision-based detection methods still have unstable issue. Authors can further explain and clarify this.

According to the results, indeed the thermal detector performs better due to thermal information stronger discriminative characteristics. However, that only happens for controlled temperature environments. While operating UAVs e.g., in hot and sunny days, thermal information becomes worthless, hence other modalities must complement this limitation. Proper discussion is addressed after Table 2: "All detectors achieve outstanding generalizable behaviour. Comparatively, the thermal detector stands out slightly. The thermal information, specifically,  discriminates the object better which facilitates detection. Apart from that, thermographic information alone could suffer from particular extreme heat weather conditions (e.g., hot and sunny days). Thus, visual and point cloud information can complement this limitation. Decently, the early fusion detector nearly matches the fine-tuned detectors. "

Multimodal systems enable modalities to complement each other to overcome individual limitations. Corresponding rephrasing is performed in the Introduction section: "For example thermal cameras discriminate really well different objects emitting different temperature intensities, however on extreme heated environments (e.g., sunny days, hot rooms) the background and foreground becomes indistinguishable and homogeneous. Visual cameras provide colour and texture information, but for foggy, glary or dark environments the signal becomes corrupted and noisy. LiDAR sensors extract depth information by analyzing light energy reflected from objects in the surroundings, nevertheless in rainy and foggy environments the reflected information becomes worthless."

3) in the weather testing, it would be better to test other weather conditions, such as rainy conditions, I hate to ask this though. If it is difficult, authors are suggested to discuss this or leave it to future work.

Rainy conditions are not addressed simply because it is not feasible for current UAV technology. A proper rephrasing is done in the Testing section as follows:

"Glary, dark and foggy weather conditions are addressed. Particularly, it is not feasible to operate current UAV technology in rainy conditions, hence rain settings are not addressed. Also, it is inopportune to apply LiDAR technology for rainy conditions [39] Heinzler, R., Piewak, F., Schindler, P., & Stork, W. (2020). Cnn-based lidar point cloud de-noising in adverse weather. IEEE Robotics and Automation Letters, 5(2), 2514-2521..."

4) the last concern is about the designed marker. I am curious about this specific design. Authors may further explain this. Is it possible to replace this marker with some ordinary objects? Ref: [1] Zhang, H., Sun, M., Li, Q., Liu, L., Liu, M. and Ji, Y., 2021. An empirical study of multi-scale object detection in high resolution UAV images. Neurocomputing, 421, pp.173-182.

Thank you for the question. It is not possible to replace the marker with an ordinary object because the used marker provides spatially aligned multimodal information that enables precise and robust detection against diverse weather conditions. Other way, by using single modality, the detection is vulnerable against specific sensory limitations. This concern is addressed in the end of the introduction where you may find the following rephrasing: "This combination of sensors is implemented by a multimodal fiducial ArUco marker called ArTuga. It enables multimodal detection against several weather conditions for robotic solutions endowing heterogeneous perception systems comprised by visual, thermographic and LiDAR based devices. Therefore, it is not possible to replace the marker with an ordinary object given the constructive characteristics of the ArTuga that provide a spatial alignment of certain elements that facilitate the precise and robust detection of the marker."

Round 2

Reviewer 1 Report

Authors have incorporated my concerns satisfactorily.